# Association of Food and Alcohol Consumption with Peripheral Atherosclerotic Plaque Volume as Measured by 3D-Ultrasound

**DOI:** 10.3390/nu12123711

**Published:** 2020-11-30

**Authors:** Maria Noflatscher, Michael Schreinlechner, Philip Sommer, Philipp Deutinger, Markus Theurl, Rudolf Kirchmair, Axel Bauer, Peter Marschang

**Affiliations:** 1Department of Internal Medicine III (Cardiology, Angiology), Medical University of Innsbruck, Anichstr. 35, A-6020 Innsbruck, Austria; Michael.Schreinlechner@i-med.ac.at (M.S.); philip.sommer@tirol-kliniken.at (P.S.); Philipp.Deutinger@student.i-med.ac.at (P.D.); Markus.Theurl@i-med.ac.at (M.T.); Rudolf.Kirchmair@i-med.ac.at (R.K.); Axel.Bauer@i-med.ac.at (A.B.); Peter.Marschang@i-med.ac.at (P.M.); 2Department of Internal Medicine, Via Lorenz Boehler, 5, Central Hospital of Bolzano (SABES-ASDAA), I-39100 Bolzano-Bozen, Italy

**Keywords:** atherosclerosis, 3D-ultrasonography, nutrition, alcohol consumption, vascular diseases

## Abstract

Background: Food patterns and alcohol consumption influence the risk for cardiovascular diseases (CVD) and a healthy nutrition is essential for the prevention of CVD. The aim of this study was to determine the influence of nutrition and alcohol consumption on peripheral atherosclerotic plaque volume (PV) using an innovative 3D ultrasound approach. Methods: In this prospective, single centre study we included 342 patients with at least one cardiovascular risk factor or established CVD. PV in the carotid and femoral artery was measured using a semi-automatic software. Information on food and alcohol consumption of the participants was collected using an internationally acknowledged standardized questionnaire (DEGS1). Results: Patients with low total PV consumed significantly more vegetables (*p* = 0.004) and vegetable juice (*p* = 0.019) per week compared to patients with high total PV. In contrast, patients with high total PV reported a higher alcohol consumption compared to patients with low total PV (*p* = 0.026). Patients without vascular disease, in particular cerebrovascular disease (*p* = 0.001) and peripheral arterial disease (*p* = 0.012), reported a significantly higher fish consumption per week. In the multivariate model, we found a significant negative association for vegetable consumption (*p* = 0.034) and female gender (*p* = 0.018) but a significant positive association for alcohol (*p* = 0.001), age (*p* < 0.001) the presence of vascular disease (*p* < 0.001) and cardiovascular risk factors (*p* < 0.001) with total PV. Conclusion: In this study we were able to show an association of food and alcohol consumption with peripheral atherosclerotic PV measured by 3D-ultrasonography. Following a healthy nutritional lifestyle (vegetable consumption, no excessive alcohol consumption) and regular fish consumption appears to be associated with less peripheral atherosclerosis and decreased prevalence of vascular diseases, respectively.

## 1. Introduction

Cardiovascular diseases (CVD) represent the leading cause of death in high-income countries, although mortality rates have decreased over the last decades. Food patterns influence the risk for CVD [1,2,3], and a healthy nutrition is essential for the prevention of CVD [1,2,3]. These associations may be explained by the fact that food patterns affect cardiovascular risk factors (CVRF) like cholesterol, body weight, blood pressure and diabetes mellitus [4]. An association between a healthy diet and the risk for CVD has been reported for elderly men and women without previous CVD [5]. Dehghan et al. have shown in their study including more than 30,000 patients with known vascular disease or diabetes that a healthy nutrition reduces the incidence for recurrent CVD. In this trial, the benefit of a healthy diet was seen in addition to efficient drug therapy for secondary prevention [6].

Alcohol consumption is the fifth leading risk factor for death and inability to work worldwide [7]. The association of alcohol and CVD is complex showing a J-shaped curve with increasing intake and appears to depend also on drinking habits. Several observational studies have shown that light to moderate alcohol consumption (1–2 units per day) reduces cardiovascular risk [8,9], whereas heavy consumption increases cardiovascular risk [10]. Similarly, a recent study that divided the participants into six drinking categories (lifelong abstinence, lifelong occasional drinker, former drinker, light, moderate and heavy drinker) showed that low to moderate alcohol consumption reduces all cause and CVD-specific mortality in U.S. adults, whereas heavy drinking increases all-cause and cancer-specific mortality [11]. In contrast to moderate wine and beer consumption, the intake of spirit appears to be associated with an increased risk for coronary heart disease (CHD) [12]. Moreover, binge drinking and drinking outside mealtimes increases the risk, whereas drinking during mealtimes and on several days of the week is associated with the lowest health risk [13]. Ronsley et al. described in their meta-analysis a relative risk reduction of 25% for cardiovascular mortality and 13% for total mortality in alcohol consumption compared to participants drinking no alcohol. The greatest risk reduction was seen between 12.5 g and 25 g alcohol per day (150–300 mL wine) [9].

Sillesen and colleagues showed that peripheral arteriosclerotic plaque burden strongly correlates with the coronary artery calcium score. They were also the first to use semi-automatic software for the exact quantification of atherosclerotic plaque burden in peripheral arteries [14]. Shortly thereafter, a system with a rotating three-dimensional (3D) ultrasound probe and plaque quantification software became commercially available and has proven to be a promising new method for the non-invasive quantification of peripheral atherosclerosis [15]. Ultrasound is a broadly available technique without the need for radiation or contrast medium [16]. Interestingly, the distribution of plaque volume (PV) appears to depend on the bifurcation angle, with large angles being associated with higher PV [17].

The aim of this study was to determine the influence of nutrition and alcohol consumption on peripheral atherosclerotic PV using an innovative 3D ultrasound approach.

## 2. Methods

### 2.1. Study Design

The study “Correlation of Atherosclerotic PV and Intima Media Thickness with Soluble P-selectin” (ClinicalTrials.gov Identifier: NCT01895725) is a prospective, observational single center cohort study with participants between 30 and 85 years of age. The baseline results of the study have been described in detail previously [18]. All patients presenting to the outpatient clinic at the Department of Internal Medicine III (cardiology, angiology) of Innsbruck Medical University with routine indications for ultrasound examinations of the carotid and/or femoral arteries were screened for potential inclusion into the study. Men and women with established coronary artery disease (CAD), cerebrovascular disease (CBVD) or peripheral arterial disease (PAD) or at least one traditional CVRF (arterial hypertension, smoking, hyperlipidemia, diabetes or family history of CVD) were included. Between May 2018 and May 2019, each of the study participants received a questionnaire, 342 of which were completed (77% response rate—See Figure 1).

Positive family history was defined as premature CVD in a first degree relative (<55 years in men and <65 years in females). For the diagnosis of diabetes, a fasting glucose level ≥126 mg/dL or the use of diabetes medication was considered relevant. Arterial hypertension was defined as systolic blood pressure ≥140 mmHg and/or diastolic blood pressure ≥90 mmHg and/or current antihypertensive therapy. For hyperlipidemia a low-density lipoprotein value ≥160 mg/dL and/or triglycerides ≥150 mg/dL and/or the use of lipid lowering drugs were required. The estimated glomerular filtration rate (eGFR) was calculated by the Modification of Diet in Renal Disease (MDRD) formula, and chronic kidney disease (CKD) was defined as eGFR <60 mL/min/1.73 m^2^.

The study protocol has been approved by the Ethics Committee of the Medical University of Innsbruck and complies with the Declaration of Helsinki. Written informed consent was provided by all participants before inclusion into the study.

### 2.2. Ultrasound Imaging

Ultrasound imaging of the carotid and femoral artery including intima media thickness (IMT) measurement and 3D plaque volumetry was performed, as described previously using a Philips iU22 system (Philips, Amsterdam, The Netherlands) [18]. For the IMT measurements, we used a linear L9-3 probe and a built-in, automatic mean IMT calculation software. IMT measurements were performed electrocardiogram-triggered in end-diastole (as determined by the R wave) in the far wall of the distal common carotid artery or the proximal superficial femoral artery 1 cm distal to the flow divider along a segment of 10 mm free of plaques. Plaques were defined as local structures extending at least 0.5 mm into the arterial lumen, or 50% of the surrounding IMT, or showing a thickness >1.5 mm, as measured from the media–adventitia interface to the intima–lumen interface according to the Mannheim consensus [19]. Plaque volumetry was performed using the Philips iU22 ultrasound system equipped with a VL13-5 mechanical 3D probe and a semi-automatic plaque quantification software (QLAB). Using this software, the vessel was divided into numerous cross sections over a distance of 6 cm including the bifurcation and the adjacent parts of the internal and common carotid or common femoral and superficial femoral artery, respectively. After optimization of the ultrasound image settings, the outline of the vessels was defined above and below the bifurcation, and a key image with definition of the plaque borders was generated. The software then calculated the plaque volume automatically. When necessary, to correctly display plaque structures, the sensitivity of the automatic plaque detection was modified by setting the sensitivity level manually. Carotid and femoral PV was defined as the sum of PV of both sides, respectively. The sum of carotid and femoral PV was defined as total PV. Assessment of inter-observer variability of three different observers revealed very good agreement between the raters with an intra-class correlation coefficient of 0.95 (95% CI, 0.82–0.99).

### 2.3. Dietary Assessment

Information on the food consumption of the participants was collected using an internationally acknowledged standardized questionnaire, named DEGS1 (questionnaire for “Studie zur Gesundheit Erwachsener in Deutschland”). This questionnaire was developed by the Robert Koch Institute for the health evaluation of adults in Germany [20]. The questionnaire collects portion size and frequency of different food groups, thereby collecting the consumption of 53 foods and food groups in the last 4 weeks. The questionnaire was handed out during the study visit, completed by the patient and collected at the follow-up visit.

Each food item receives a specific numerical value assigned from the DEGS, which represents the overall consumption frequency of the food over the last month. This value was multiplied with the portion size and divided by 4 to calculate the average consumed portion of the food group per week. The portion size was defined for each food group by the study for the health of adults in Germany.

### 2.4. Statistical Analysis

Normal distribution was tested using the Kolmogorov and Smirnov Test [21]. Continuous variables following a normal distribution were presented as mean ± standard deviation (SD), whereas parameters not following a normal distribution were given as median and interquartile range (IQR) or mean and 95% confidence interval. Bootstrapping was performed to calculate 95% confidence intervals. Categorical variables are shown in absolute numbers and percentages. The Mann–Whitney U test was used to calculate differences of continuous variables by a categorical independent variable with two or more groups. To compare categorical variables, we applied the Chi square test. For some analyses, PV was separated into high (255–1887 mm^3^) and low (0–254 mm^3^) using a previously applied value as a cut-off [18]. The relationship between peripheral PV and food consumption was examined using linear regression. To examine the association of food consumption and other factors with peripheral atherosclerosis, at first univariate linear regression was applied (data not shown). The parameters alcohol and vegetable consumption as well as age, gender, BMI (body mass index), known vascular disease and CVRFs with a significant *p*-value < 0.05 in the univariate linear regression were then used in the multivariable regression model. A *p*-value of <0.05 was considered as significant. Spearman’s test was applied to evaluate the linear correlations for selected variables. SPSS Statistic was used for all statistical analysis (version 24.0; IBM Corp, Armonk, NY, USA).

## 3. Results

### 3.1. Characteristics of Study Population

A total of 342 patients (42% female) with a mean age of 66 years, ranging between 58 and 73 years, completed the food questionnaires. Baseline characteristics, including demographic, clinical and laboratory characteristics, are summarized in Table 1. Patients are divided into low and high total plaque volume using a cut-off of 255 mm^3^, as described previously [18].

The most common CVRF was hyperlipidemia (89.2%) followed by hypertension (67.5%). About a quarter of the study population (24.6%) had a positive family history for CVD and 12.9% suffered from diabetes mellitus. More than half of the patients were under lipid lowering (57.8%) and antihypertensive (61.1%) therapy.

All 342 participants underwent 3D ultrasonography to measure PV in the carotid and femoral arteries.

### 3.2. Distribution of Atherosclerotic PV According to the Consumption of Vegetables and Fruits

We observed a significant difference in the consumption of vegetables between the high and the low total PV group. Patients with low total PV consumed significantly more vegetables and vegetable juice per week. Conversely, there was no significant difference in the consumption of fruits and fruits juice between the two groups, as shown in Table 2.

Vegetable consumption showed a negative correlation with total PV in the overall collective (r = −0.194, *p* < 0.001). Vegetable juice consumption per week demonstrated a negative correlation with total PV (r= −0.135, *p* = 0.015), whereas no correlation was found for fruits and fruits juice consumption (data not shown).

### 3.3. Association of Atherosclerotic PV According to Alcohol Consumption

Patients with high total PV reported a higher total alcohol consumption (beer, wine and spirits consumption) compared to patients with low total PV (*p* = 0.026). Beer consumption was significantly (0.025) higher in the group with high total PV, whereas there was no significant difference in the consumption of wine and spirits per week between the two groups (Table 3).

We did not observe a significant association of any other food item or food group with total PV.

Alcohol consumption showed a positive correlation with total PV in the overall collective (r = 0.135, *p* = 0.016). Beer consumption per week demonstrated a positive correlation with total PV (r= 0.156, *p* = 0.004), whereas no correlation was found for wine and spirits (data not shown).

### 3.4. Association of Fish Consumption with the Presence of Vascular Diseases

Patients without vascular disease reported a significantly higher fish consumption per day compared to patient with known vascular disease (*p* = 0.043), in particular with CBVD (*p* = 0.001) and PAD (*p* = 0.012) (Figure 2). The association of fish consumption with total PV was not significant.

We did not find a significant difference in fish, alcohol, and vegetable and fruit consumption in participants with CAD compared to those without CAD.

### 3.5. Multivariate Analysis

Univariable linear analysis for total PV revealed statistically significant *p*-values for total alcohol consumption, vegetables consumption, gender, age, BMI, known vascular disease and CVRFs.

In the multivariate model, we found a significant negative association for vegetable consumption and female gender as well as a significant positive association for alcohol consumption, age, the presence of vascular disease and CVRFs with total PV but no association for BMI (see Table 4).

## 4. Discussion

In this prospective, single-center study, we examined the association of food consumption with peripheral atherosclerosis. We found that participants with low total PV, as measured by 3D-volumetry in the carotid and femoral arteries, reported significant less alcohol, in particular significant less beer consumption. In addition, patients with low total PV ate significantly more vegetables compared to patients with high total PV. Moreover, participants without pre-existing vascular disease as CBVD or PAD reported a significantly higher fish consumption.

This appears to be the first study investigating the association of food consumption with peripheral atherosclerotic PV using this innovative 3D ultrasound approach.

Already, the Northern Manhattan study from 2014 reported that a Mediterranean-style diet was associated with lower atherosclerotic plaque burden in the carotid arteries. This diet includes an abundant consumption of fruits, vegetables, monounsaturated fat, fish, whole grains, legumes and nuts; moderate alcohol consumption; and a low intake of red meat, saturated fat and refined grains [22]. Previous studies have shown that the Mediterranean diet may be associated with improved blood lipids [23], lower blood pressure [24], less obesity [25], lower insulin resistance [26] and reduced parameters of inflammation like hsCRP [27] and IL-6 [28]. Among other ingredients, organic sulphur compounds inhibit vascular inflammation as well as platelet function and lower the concentrations of plasma lipids [29]. Vegetables are a very heterogeneous food group with a complex composition. Therefore, other compounds like pectins and polyphenols have also been implicated to mediate positive effects. Polyphenols inhibit the production of reactive oxygen species, the production of superoxide and the proliferation and migration of vascular smooth muscle cells while decreasing platelet aggregation and mitochondrial oxidative stress [30].

The Dutch food-based dietary guidelines published in 2015 state that fruit and vegetable consumption is associated with a lower risk of CHD, stroke, diabetes and colorectal cancer and goes along with a reduction in blood pressure [31]. In addition, fruits and vegetables have the ability to simulate the immune system, reduce platelet aggregation and to modulate cholesterol synthesis [32]. Moderate alcohol consumption and high intake of legumes and fish were associated with a decreased risk of vascular death [33]. These beneficial effects of the increased consumption of vegetables may explain some of the findings of our study.

With regard to a Mediterranean diet, we observed an association of fish consumption with pre-existing vascular disease, particular CBVD and PAD, in our study. The American Heart Association recommends fish consumption in patients with CHD, former cardiovascular events, heart failure and reduced ejection fraction [34]. The consumption of one fish dish per week reduces the risk of fatal CHD and lowers the risk of stroke [31]. Omega 3 polyunsaturated fatty acids are known to lower triglyceride levels, inhibit platelet aggregation, lower blood pressure and have an antiarrhythmic effect [35]. Like the study by Johnsen et al. [36], we found no positive effect of fish consumption on atherosclerotic plaque burden. Additionally, another study group showed that omega 3 polyunsaturated fatty acids did not slow the progression of atherosclerosis in carotid arteries examined by ultrasound [37], whereas Buscemi et al. demonstrated an association of high fish consumption (>two servings a week) with less carotid atherosclerosis [38].

Alcohol intake and especially beer consumption was associated with a higher peripheral PV in our study. We have to mention, though, that the patients in our study reported a relatively low alcohol consumption. Several studies describe a J-shaped curve between alcohol consumption and total mortality. The lowest risk is observed with 2–4 drinks per day for men and 1–2 drinks per day for women [39]. Higher alcohol consumption is associated with increased mortality rates, hypertension, alcoholic cardiomyopathy, cancer and cardiovascular events [40]. In contrast, light to moderate alcohol consumption is associated with a lower risk of CHD and stroke [41,42,43,44] as well as lower cardiovascular and total mortality [45]. Costanzo et al. could show a J-shaped curve not only for wine but also for beer consumption, but not for spirit consumption [12]. This meta-analysis also described that wine drinkers adhere to a healthier way of life compared to beer drinkers, and that wine consumption reduces total mortality [13]. In contrast, we did not observe a difference in wine consumption among patients with high and low PV in our study.

The cardio-protective effect of alcohol has been explained by the elevation of HDL-cholesterol and adiponectin, increased insulin sensitivity and the reduction of plasma viscosity and fibrinogen. In addition, light to moderate alcohol intake increases fibrinolysis, decreases platelet aggregation, improves endothelial function, reduces inflammation and promotes antioxidant effects [40,46,47]. These effects may be explained by the polyphenolic and phenolic particles of wine, specially red wine [48] and beer [49,50], although beer and wine contain different substances [12]. Ellison et al. did not find an association of increasing alcohol intake with calcified atherosclerotic plaques in the coronary arteries and the aorta [51]. Moreover, alcohol consumption appears not to be associated with plaque thickness [52,53]. In our study, participants reporting higher alcohol—particularly beer—consumption had higher plaque burden compared to those drinking less alcohol. This effect was surprising, since most study participants consumed small amounts of alcohol. Presently, we cannot rule out alcohol and beer consumption as a marker for a less healthy lifestyle in our study cohort.

## 5. Limitations

The design of our study includes several limitations. The sample size was relatively small and the study was performed at one single center. Although our study used self-reporting of patients using standardized questionnaires, we cannot exclude under- or over-reporting of dietary habits by the patients. We have to mention that the higher vegetable consumption could also be representative of a healthier lifestyle with more physical activity, less weight and less tobacco consumption with known impacts on cardiovascular morbidity and mortality [54,55]. Our study design led to the inclusion of patients with different CVRF and different CVD. However, we could thereby show the feasibility of plaque quantification by 3-D-ultrasound in this heterogeneous group of patients and correlate peripheral atherosclerosis with their dietary habits.

## 6. Perspective

In this study, we demonstrate the association of food and alcohol consumption with peripheral atherosclerotic PV. People should try to adhere to a healthy diet to prevent the formation of atherosclerotic plaques. Larger studies with the registration of cardiovascular endpoints will be necessary to confirm these results.

## Figures and Tables

**Figure 1 nutrients-12-03711-f001:**
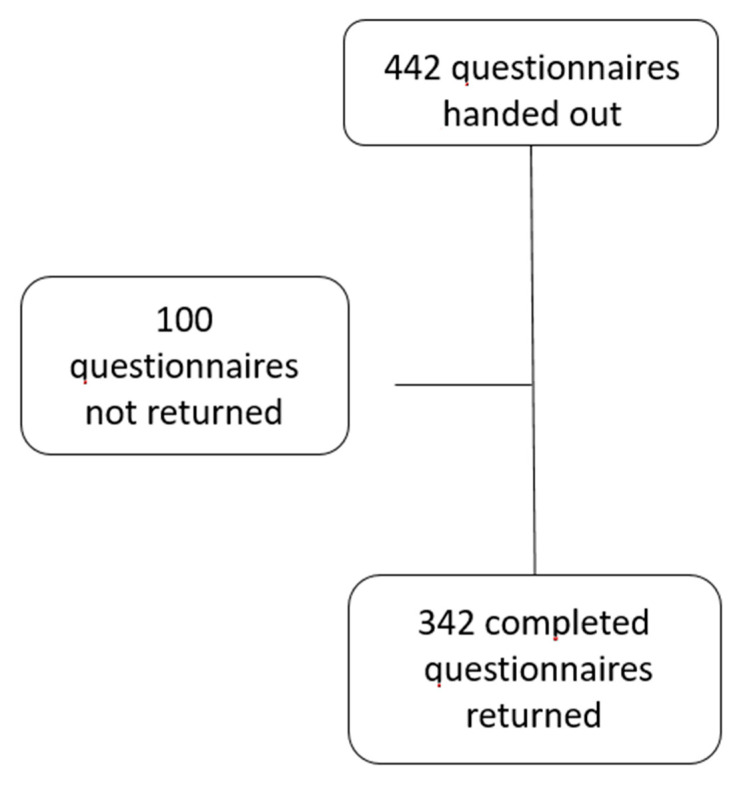
Flow chart of study participants.

**Figure 2 nutrients-12-03711-f002:**
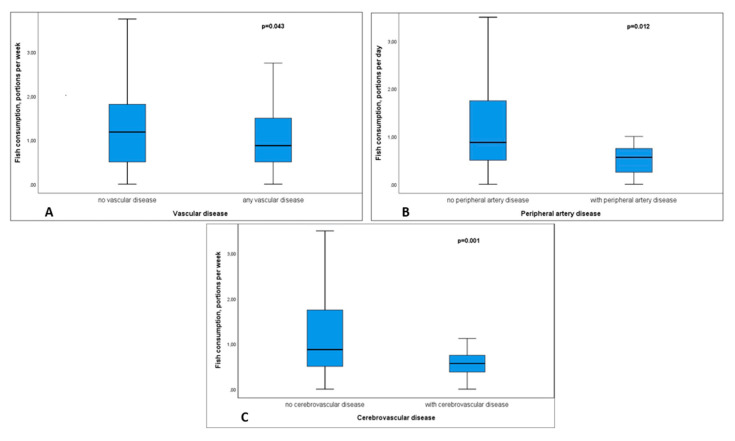
Boxplot diagram displaying differences in fish consumption per week between patients with and without vascular disease (**A**), peripheral artery disease (**B**) and cerebrovascular disease (**C**). The boxes show the median and interquartile range, while the whiskers are representative of the 95% confidence interval.

**Table 1 nutrients-12-03711-t001:** Characteristic of the study population.

	Total Population (*n* = 342)	Low Total Plaque Burden (*n* = 171, 50%, 0–254 mm^3^)	High Total Plaque Burden (*n* = 171, 50%, 255–1887 mm^3^)	*p*-Value
Age, years	66 (58–73)	61 (55–68)	69 (61–74)	**<0.001**
Female, *n* (%)	145 (42.4)	88 (51.5)	57 (33.3)	**0.001**
Body mass index, kg/m^2^	25.7 (23.7–28.4)	25.4 (23.3–28.3)	26 (24.1–28.4)	0.215
Hypertension, *n* (%)	231 (67.5)	94 (55)	137 (80.1)	**<0.001**
Family history for CVD, *n* (%)	84 (24.6)	47 (27.5)	37 (21.6)	0.210
Smoking (pack years)	20 (10–40)	15 (5–30)	30 (15–40)	**0.001**
Hyperlipidaemia, *n* (%)	305 (89.2)	148 (86.5)	157 (91.8)	0.118
Diabetes mellitus, *n* (%)	44 (12.9)	18 (10.5)	26 (15.2)	0.197
HbA1c, %	5.5 (5.4–5.8)	5.5 (5.3–5.7)	5.5 (5.4–5.9)	0.070
hs-CRP, mg/dL	0.19 (0.09–0.43)	0.17 (0.09–0.42)	0.21 (0.1–0.45)	0.354
Total cholesterol, mg/dL	192.9 (±45.3)	201.4 (±46.1)	184.3 (±42.9)	**<0.001**
LDL-cholesterol, mg/dL	114 (88.8–143)	122 (98–147.3)	106 (85–135)	**0.001**
HDL-cholesterol, mg/dL	57 (46–71)	58.5 (48–74.3)	55.5 (45–67)	**0.048**
Triglyceride mg/dL	132 (93.8–185)	126.5 (85–174.3)	135 (97–196)	0.225
Creatinin, mg/dL	0.95 (0.83–1.1)	0.92 (0.83–1.05)	0.96 (0.84–1.12)	**0.028**
Anticoagulation therapy, n, %	22 (6.4)	6 (3.5)	16 (9.4)	**0.027**
Antiplatelet therapy, n, %	163 (47.7)	61 (35.7)	102 (59.6)	**<0.001**
Lipid lowering therapy	197 (57.6)	87 (50.9)	110 (64.3)	**0.014**
Antihypertensive therapy	209 (61.1)	89 (52)	120 (70.2)	**0.001**
Any vascular disease, *n* (%)	139 (40.6)	51 (29.8)	88 (51.5)	**<0.001**
CAD, *n* (%)	119 (34.8)	42 (24.6)	77 (45)	**<0.001**
CBVD, *n* (%)	32 (9.4)	11 (6.4)	21 (12.3)	0.064
PAD, *n* (%)	19 (5.6)	4 (2.3)	15 (8.8)	**0.010**
Total plaque volume, mm^3^	254 (94.8–502.8)	95 (23–171)	502 (360–745)	**<0.001**
Femoral plaque volume, mm^3^	135.1 (32.5–281)	47 (0-101)	281 (171–478)	**<0.001**
Carotid plaque volume, mm^3^	88.5 (15–223.3)	22 (0–63)	222 (106–385)	**<0.001**
Femoral IMT, mm	0.49 (0.44–0.54)	0.47 (0.42–0.53)	0.50 (0.45–0.55)	**0.001**
Carotid IMT, mm	0.72 (0.63–0.82)	0.69 (0.61–0.79)	0.73 (0.67–0.85)	**0.001**
Systolic blood pressure, mmHg	119 (108–133)	117 (106–128)	122 (110–138)	**0.006**

Parameters are median (interquartile range) or mean (± standard deviation) as indicated for continuous variables or number (percentage) for categorical variables. Statistically significant differences (*p* < 0.05) between high and low total plaque volume are shown in bold. CVD = cardiovascular disease, LDL = low density lipoprotein, HDL = high density lipoprotein, hsCRP = high-sensitive C-reactive protein, CAD = coronary artery disease, CBVD = cerebrovascular disease, PAD = peripheral arterial disease, IMT = intima-media thickness.

**Table 2 nutrients-12-03711-t002:** Distribution of total plaque volume depending on the consumption of vegetables and fruits.

	Total Population	Low Total Plaque Volume	High Total Plaque Volume	*p*-Value
Vegetables, portions per week	6.33 (5.81–6.94)	7.18 (6.40–8.06)	5.46 (4.81–6.17)	0.004
Vegetable juice, portions per week	0.34 (0.15–0.64)	0.54 (0.19–1.28)	0.13 (0.07–0.21)	0.019
Fruits, portions per week	9.20 (8.18–10.22)	9.30 (7.87–10.93)	9.10 (7.73–10.73)	0.848
Fruits juice, portions per week	3.50 (2.73–4.34)	2.81 (2.06–3.73)	4.18 (3.03–5.46)	0.282

Portions are 150 g for vegetables and fruits and 200 mL for vegetable and fruit juice, respectively. Parameters are median (95% confidence interval) as indicated. Statistical significant differences (*p* < 0.05) between high and low total plaque volume are shown in bold.

**Table 3 nutrients-12-03711-t003:** Association of total plaque volume depending on alcohol consumption.

	Total Population	Low Total Plaque Volume	High Total Plaque Volume	*p*-Value
Total alcohol, drinks per week	5.04 (4.22–5.87)	3.57 (2.78–4.35)	6.51 (5.27–7.88)	0.026
Beer, drinks per week	2.12 (1.64–2.68)	1.53 (1.02–2.12)	2.73 (1.91–3.69)	0.025
Wine, drinks per week	2.52 (2.06–3.00)	1.79 (1.39–2.16)	3.27 (2.38–4.16)	0.497
Spirits, drinks per week	0.30 (0.21–0.41)	0.19 (0.13–0.26)	0.39 (0.25–0.61)	0.151

Alcohol consumption is given as drinks/week defined as 330 mL for beer, 125 mL for wine, and 2 cl for spirits, respectively; Parameters are mean (95% confidence interval) as indicated Statistical significant differences (*p* < 0.05) between high and low total plaque volume are shown in bold.

**Table 4 nutrients-12-03711-t004:** Multivariate prediction model for total plaque volume.

	Total Plaque Volume (mm^3^)	
Parameter	B (95CI)	*p*-Value
Total alcohol consumption (drinks per week)	8.67 (3.60–13.74)	**0.001**
Vegetables consumption (portions per week)	−7.49 (−14.42–−0.57)	**0.034**
Age (years)	12.83 (8.90–16.77)	**<0.001**
Female gender	−93.98 (−171.78–−16.18)	**0.018**
BMI (kg/m^2^)	4.56 (−3.66–12.78)	0.276
Vascular disease	141.82 (65.24–218.4)	**<0.001**
CVRF	81.59 (41.60–121.59)	**<0.001**

The regression coefficients for total plaque volume are shown. Statistically significant regression coefficients are shown in bold. Vegetables portions per week are 150 g, respectively. Vascular disease was defined as history of coronary artery disease, cerebrovascular disease or peripheral arterial disease. B = regression coefficient, CI = confidence interval, BMI = body mass index; CVRF = cardiovascular risk factor (arterial hypertension, smoking, hyperlipidemia, diabetes or family history of cardiovascular disease).

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
