# Peer review of "Association of Food and Alcohol Consumption with Peripheral Atherosclerotic Plaque Volume as Measured by 3D-Ultrasound"

_nutrients, 2020, doi:10.3390/nu12123711_

Round 1
Reviewer 1 Report
This manuscript focuses on interesting topic of non-invasive atherosclerosis imaging in patients with vascular atherosclerotic diseases or cardiovascular risk factor and the association with food and alcohol consumption.
However, there are some major limitations and I suggest some additional discussion of the following comments:
- The selection procedure of patients with CAD, CBVD, PAD or at least one traditional CVRF is not well described. This should be clearly described. There is also a great heterogenicity of patients with established cardiovascular disease (secondary prevention) and only cardiovascular risk factors (primary prevention).
- The description of ultrasound imaging is absolutely insufficient as this is the main topic of the paper:
- More information on the ultrasound probe (matrix probe, mechanical 3D probe, ultrasound system used) are needed.
- More information on the ultrasound imaging settings (gain, dynamic range, etc.)
- 3D-analysis of plaque volume measurement should be described in much more detail. 3D-measurement of plaque are very difficult and error prone. How the researcher measured plaque volume in calcified plaque with ultrasound shadowing. There is also not clear how the researchers defined lumen – plaque boarder in hypoechoic plaques? It is not clear how long the segment of the plaque was defined for the plaque volume (1cm or 6cm)? Please provide also more information on the method used for plaque segmentation and volume measurement.
- The group of low and high PV seems quite arbitrary. As PV is continuous value the correlation with food and alcohol consumption could be also by done by using continuous values of PV and not categorial.
- For the multivariate analysis all significant factors should be included ant not only age and gender (Hypertension, Smoking, Diabetes, Lipid values, Creatinine and vascular diseases)
- The results of fish consumption and the presence of vascular diseases was not the main focus of the study described in the method. This seems very confusing as there are no relation to the 3D ultrasound measurement and plaque volume.
Author Response
Reviewer 1:
1. The selection procedure of patients with CAD, CBVD, PAD or at least one traditional CVRF is not well described. This should be clearly described. There is also a great heterogenicity of patients with established cardiovascular disease (secondary prevention) and only cardiovascular risk factors (primary prevention).
All patients presenting to the outpatient clinic of our department with routine indications for ultrasound examinations of the carotid and/or femoral arteries were screened for potential inclusion into the study. Men and women with established coronary artery disease (CAD), cerebrovascular disease (CBVD) or peripheral arterial disease (PAD) or at least one traditional CVRF (arterial hypertension, smoking, hyperlipidaemia, diabetes or family history of CVD) were included. The selection procedure of patients has been added to the methods section (page 5) of the revised manuscript.
This design led to the inclusion of patients with different CVRF and different CVD. However, we could thereby show the feasibility of plaque quantification by 3-D-ultrasound in this heterogeneous group of patients and correlate peripheral atherosclerosis with their dietary habits. The heterogenicity of the study population has been added to the limitation factors in the discussion.
2. The description of ultrasound imaging is absolutely insufficient as this is the main topic of the paper:
More information on the ultrasound probe (matrix probe, mechanical 3D probe, ultrasound system used) are needed.
More information on the ultrasound imaging settings (gain, dynamic range, etc.)
3D-analysis of plaque volume measurement should be described in much more detail. 3D-measurement of plaque are very difficult and error prone. How the researcher measured plaque volume in calcified plaque with ultrasound shadowing. There is also not clear how the researchers defined lumen – plaque boarder in hypoechoic plaques? It is not clear how long the segment of the plaque was defined for the plaque volume (1cm or 6cm)? Please provide also more information on the method used for plaque segmentation and volume measurement.
The description of ultrasound imaging, as well as information’s about the ultrasound probes used and the ultrasound imaging settings have been added to the methods section (ultrasound imaging) of the manuscript. Especially the 3D-ultrasound measurement of plaque volume has been described in much more detail. Using the built-in semi-automated software, the vessel is divided into numerous cross sections over a distance of 6 cm including the bifurcation and the adjacent parts of the internal and common carotid or common femoral and superficial femoral artery, respectively. Before assessing the plaque volume, ultrasound image settings like gain, filters and dynamic range were optimized by pressing the iSCAN button. The outline of the vessels was defined above and below the bifurcation and a key image with definition of the plaque borders was generated. The software then calculates the plaque volume automatically. In our hands, ultrasound shadowing by calcified plaques did not appear to be a limiting factor for these measurements. In the presence of hypoechoic plaques, however, careful visualisation with a linear ultrasound probe using a sensitive colour mode (Power Doppler) was performed before plaque volumetry. With this information, the sensitivity of the automatic plaque detection can then be modified by setting the sensitivity level manually.
3. The group of low and high PV seems quite arbitrary. As PV is continuous value the correlation with food and alcohol consumption could be also by done by using continuous values of PV and not categorical.
Patients were divided into two groups (low and high PV) to facilitate data presentation and to be consistent with our previous publication (1). Spearman’s test was applied to evaluate the linear correlations for selected variables (see results section). As suggested by the reviewer, we changed the binary logistic regression model into a multivariate linear regression model (table 4).
4. For the multivariate analysis all significant factors should be included ant not only age and gender (Hypertension, Smoking, Diabetes, Lipid values, Creatinine and vascular diseases)
We included CVRF and vascular diseases in the multivariate analysis (Table 4). Since creatinine was not significant in the univariate analysis, we did not include it in the multivariate analysis.
5. The results of fish consumption and the presence of vascular diseases was not the main focus of the study described in the method. This seems very confusing as there are no relation to the 3D ultrasound measurement and plaque volume.
We agree with the reviewer, that this association is not related to the main focus of our study. We think, however, that this observation is nevertheless important and worth to be reported. To avoid confusion, we combined the original figures 2 – 3 into one figure (new figure 2). In addition, the lack of an association between fish consumption and total plaque volume is now stated explicitly in the results section (page 13).

Reviewer 2 Report
This paper used standard statistical methods to assess differences between two groups, low and high PV. I am not familiar with some of the variables, some forgive my ignorance is some answers to some of the queries below are obvious. But there is some clarifications need in regards to the statistics results reported.
Given that PV is a continuous measure, why dichotomize into two groups. I would have thought that a lot of important information would be lost, although perhaps there is a good reason why this was done. If that is the case, then it would good to make that clear. I don't skew is necessarily a good reason, since transformations can account for this in regression analyses and there are non-parametric options too. Other than considering why not using regression analyses, why was the median chosen? Is there no understanding of what levels define high and low? Also, if categorizing is necessary, why just two groups. More information is retained by splitting into more than two groups, and the sample size will be okay for that. A brief discussion that may be useful is "The cost of dichotomizing continuous variables" by Altman and Royston (2006), BMJ.
Where was the Kruskal Wallis test used? I couldn't see where it would have been.
Table 1: I think the results need to be checked. Checking, e.g., CBVD with a chisquare test with and without correction I obtained a very different p-value comparing the groups.
Table 2 and Table 3: I have a couple of important points here. Please clarify how the p-values comparing the groups were computed. If this was using a two sample comparison of means (which would be the natural choice given that means and confidence intervals are reported for the total and the groups), then these p-values should be checked. Even if the Bonferroni correction was used for these (if that is the case then please make that clear), then even then some appear way too large. For example, based on the CIs for the individual groups, the p-value in the first row of Table 3 should be much smaller than the 0.026 reported. Additionally, if this is based on comparison of means, please report the intervals for all difference in means (this will help in understanding clinical significance, not just statistical significance, and is also is much more informative than a simple p-value) and the p-values even when not significant. Overlapping confidence intervals can still indicate a significant difference in means, and I think there will be close to significance in the last row of Table 2. Also, I can't see how the last two rows of Table 3 can indicate insignificance between the groups, although perhaps the Bonferroni correction but this is not clear.
Author Response
Reviewer 2:
- Given that PV is a continuous measure, why dichotomize into two groups. I would have thought that a lot of important information would be lost, although perhaps there is a good reason why this was done. If that is the case, then it would good to make that clear. I don't skew is necessarily a good reason, since transformations can account for this in regression analyses and there are non-parametric options too. Other than considering why not using regression analyses, why was the median chosen? Is there no understanding of what levels define high and low? Also, if categorizing is necessary, why just two groups. More information is retained by splitting into more than two groups, and the sample size will be okay for that. A brief discussion that may be useful is "The cost of dichotomizing continuous variables" by Altman and Royston (2006), BMJ.
Patients were divided into two groups (low and high PV) to facilitate data presentation and to be consistent with our previous publication (1). For this reason, dichotomisation was chosen instead of dividing the study population into quartiles or quintiles. Spearman’s test was applied to evaluate the linear correlations for selected variables (see results section). As suggested by the reviewer, we changed the binary logistic regression model into a multivariate linear regression model (table 4).
- Where was the Kruskal Wallis test used? I couldn't see where it would have been.
The Kruskal Wallis test had been erroneously included in the statistical method and was not used for any of the analyses. It has now been deleted from the revised manuscript.
- Table 1: I think the results need to be checked. Checking, e.g., CBVD with a chisquare test with and without correction I obtained a very different p-value comparing the groups.
We thank the reviewer, since table 1 contained indeed an error (e.g. p-value of CBVD) that has been corrected in the revised manuscript.
- Table 2 and Table 3: I have a couple of important points here. Please clarify how the p-values comparing the groups were computed. If this was using a two sample comparison of means (which would be the natural choice given that means and confidence intervals are reported for the total and the groups), then these p-values should be checked. Even if the Bonferroni correction was used for these (if that is the case then please make that clear), then even then some appear way too large. For example, based on the CIs for the individual groups, the p-value in the first row of Table 3 should be much smaller than the 0.026 reported. Additionally, if this is based on comparison of means, please report the intervals for all difference in means (this will help in understanding clinical significance, not just statistical significance, and is also is much more informative than a simple p-value) and the p-values even when not significant. Overlapping confidence intervals can still indicate a significant difference in means, and I think there will be close to significance in the last row of Table 2. Also, I can't see how the last two rows of Table 3 can indicate insignificance between the groups, although perhaps the Bonferroni correction but this is not clear.
The p- values comparing the groups were computed using the Mann-Whitney-U-Test (vegetables, portions per week (Mann-Whitney U =11018, Wilcoxon W =24384, Z = - 2.91, p= 0.004); vegetable juice, portions per week (Mann-Whitney U= 11942.5, Wilcoxon W= 25145.5, Z= -2.35, p= 0.019); fruits, portions per week (Mann-Whitney U= 13037.5, Wilcoxon W= 25917.5, Z =-0.19, p= 0.848); fruits juice, portions per week (Mann-Whitney U= 12926.5, Wilcoxon W= 26954.5, Z = -1.08, p= 0.282); total alcohol consumption per week (Mann-Whitney U= 10607.5, Wilcoxon W= 23168.5, Z = -2.23, p= 0.026); beer consumption per week (Mann-Whitney U= 11647, Wilcoxon W= 25675, Z = -2.24, p= 0.025); wine consumption per week (Mann-Whitney U= 12874, Wilcoxon W= 26569, Z = -0.68, p= 0.497); spirits consumption per week (Mann-Whitney U= 12428, Wilcoxon W= 25794, Z = -1.44, p= 0.151). As we tested only two groups no Bonferroni correction was applied. In addition Bootstrapping was performed to calculate confidence intervals. The results in Table 2 and 3 have been checked and found to be displayed correctly.
As suggested by reviewer 2, the whole manuscript was checked for the style of English language and moderate changes have been performed throughout the manuscript.
Reference
- Noflatscher M, Schreinlechner M, Sommer P, Kerschbaum J, Berggren K, Theurl M, et al. Influence of Traditional Cardiovascular Risk Factors on Carotid and Femoral Atherosclerotic Plaque Volume as Measured by Three-Dimensional Ultrasound. J Clin Med. 2018;8(1).

Reviewer 3 Report
In the manuscript number nutrients-1000336 entitled “Association of food and alcohol consumption with peripheral atherosclerotic plaque volume as measured by 3D-ultrasound”, the authors determine the influence of nutrition and alcohol consumption on peripheral atherosclerotic plaque volume (PV) using an innovative 3D ultrasound approach. The authors highlight that a healthy nutritional lifestyle and regular fish consumption appears to be associated with less peripheral atherosclerosis and decreased prevalence of vascular diseases, respectively.
The topic is interesting and these outcomes could be useful for clinical practice. However, the issues reported below should be addressed to improve the quality of this work.
Criticisms:
- In Table 1 some parameters that would best characterize the population are lacking. The Authors must include important factors such as: diastolic and systolic blood pressure, glycaemia, IMT and PV values.
- Several factors can influence the analysis. Thus, the authors could verify if the results remain significant also after correction for therapies, hypertension, BMI, age and gender.
Author Response
Reviewer 3:
- In Table 1 some parameters that would best characterize the population are lacking. The Authors must include important factors such as: diastolic and systolic blood pressure, glycaemia, IMT and PV values.
We agree with the reviewer that these are important characteristics of the study population which have now been added to Table 1.
- Several factors can influence the analysis. Thus, the authors could verify if the results remain significant also after correction for therapies, hypertension, BMI, age and gender.
Cardiovascular risk factors and the BMI have been added to the multivariate regression model (Table 4). Since different therapies are associated with or implied in cardiovascular diseases and cardiovascular risk factors (see definition of hyperlipidemia, hypertension and diabetes in the methods section (pages 5-6)), we did not specifically included them in the multivariate regression model

Round 2
Reviewer 1 Report
The authors addressed the reviewer’s comments and changed the manuscript accordingly. The manuscript substantially improved. No further comments are required.
Author Response
Reviewer 1:
The authors addressed the reviewer’s comments and changed the manuscript accordingly. The manuscript substantially improved. No further comments are required.
We thank the reviewers for their suggestions, which have helped us to substantially improve our manuscript.
Reviewer 2 Report
Thank you for taking to the time to clarify and revise.
A couple of minor points:
Table 4 needs to be fixed as the headers still refer to Odds Ratios.
I would recommend to report non-significant p-values instead of just n.s. while 0.05 is a cutoff, small p-values not below this are usually seen as indicative of a likely difference (perhaps not powered).
Author Response
Reviewer 2:
Thank you for taking to the time to clarify and revise.
A couple of minor points:
Table 4 needs to be fixed as the headers still refer to Odds Ratios.
I would recommend to report non-significant p-values instead of just n.s. while 0.05 is a cutoff, small p-values not below this are usually seen as indicative of a likely difference (perhaps not powered).
The headers of table 4 has been changed (regression coefficient B instead of odds ratios).
Instead of “n.s.”, non-significant p-values have been added to the tables 1 – 4 of the manuscript.